# Multitasking by the OC Lineage during Bone Infection: Bone Resorption, Immune Modulation, and Microbial Niche

**DOI:** 10.3390/cells9102157

**Published:** 2020-09-24

**Authors:** Philip M. Roper, Christine Shao, Deborah J. Veis

**Affiliations:** 1Division of Bone & Mineral Diseases, Musculoskeletal Research Center, Washington University School of Medicine, Saint Louis, MO 63110, USA; proper@wustl.edu (P.M.R.); shaochristine@wustl.edu (C.S.); 2Department of Pathology and Immunology, Washington University School of Medicine, Saint Louis, MO 63110, USA; 3Shriners Hospitals for Children, Saint Louis, MO 63110, USA

**Keywords:** osteoclast, osteomyelitis, bone, *Staphylococcus aureus*, *Porphyromonas gingivalis*, neutrophil, macrophage

## Abstract

Bone infections, also known as infectious osteomyelitis, are accompanied by significant inflammation, osteolysis, and necrosis. Osteoclasts (OCs) are the bone-resorbing cells that work in concert with osteoblasts and osteocytes to properly maintain skeletal health and are well known to respond to inflammation by increasing their resorptive activity. OCs have typically been viewed merely as effectors of pathologic bone resorption, but recent evidence suggests they may play an active role in the progression of infections through direct effects on pathogens and via the immune system. This review discusses the host- and pathogen-derived factors involved in the in generation of OCs during infection, the crosstalk between OCs and immune cells, and the role of OC lineage cells in the growth and survival of pathogens, and highlights unanswered questions in the field.

## 1. Introduction

As in other sites, the presence of pathogens in bone first elicits an innate immune response in which mature neutrophils and macrophages surround the invaders to form abscesses [1]. *Staphylococcus (S.) aureus* is the leading causative agent of infectious osteomyelitis (iOM) for most of the skeleton, except for iOM of the jaw related to periodontal diseases, where *Porphyromonas (P.) gingivalis* predominates [1,2]. A variety of other pathogens, however, are also capable of infecting bone [3,4,5]. The combined actions of the pathogens and host cells lead to high levels of many inflammatory cytokines, which expand the pool of osteoclast (OC) precursors and mature bone-resorbing cells. If the infection is not cleared by the immune system or medical treatment, with time, vascular compromise in infected tissue leads to necrotic areas of bone called sequestra that serve as a niche for chronic infection and incite further osteoclastic bone resorption [6]. In many cases, despite a robust immune response, bacteria can persist within sequestra, rendering patients generally incurable with antibiotics alone and in need of surgical intervention. As the site of hematopoiesis, bone is home to stem and progenitor populations as well as mature effector cells. Thus, given the variety of cells in this microenvironment, there is an opportunity for many factors to impact iOM pathogenesis. This review focuses on OCs as a specific node of host–pathogen interaction that has largely been overlooked except as a source of tissue damage and highlights this cell’s potential to modify disease progression.

## 2. Homeostatic Osteoclastogenesis

OCs are giant, multinucleated cells tasked with resorbing old or damaged bone as part of the homeostatic turnover needed to maintain proper skeletal health. They differentiate from immature myeloid progenitors primarily due to the actions of the osteoclastogenic cytokine receptor activator of NF-κB ligand (RANKL) [7,8,9]. Under normal physiological conditions, bone-forming osteoblasts and bone-residing osteocytes provide RANKL and supportive macrophage-colony stimulating factor (M-CSF), which drives commitment to the OC lineage and induces the fusion of these pre-OCs to form mature OCs. Osteoblast lineage cells also produce soluble RANKL decoy receptor osteoprotegerin (OPG), which creates the RANK/RANKL/OPG axis that regulates osteoclastogenesis. RANKL binding its receptor RANK initiates downstream signaling that is mediated by interaction between the adaptor protein TNF-receptor-associated factor 6 (TRAF6) and the cytoplasmic domain of RANK. The master transcription factor nuclear factor of activated T cells cytoplasmic 1 (NFATc1) is activated via a combination of immunoreceptor tyrosine-based activation motif (ITAM) co-receptor signaling and downstream activation of mitogen-activated protein kinases (MAPKs), nuclear factor-κB (NF- κB), and activator protein 1 (AP-1). Early committed pre-OCs express tartrate-resistant acid phosphatase (TRAP) and αvβ5 integrin, whereas late precursors and mature OCs express cathepsin K, calcitonin receptor, TRAP5b, and αvβ3 integrin. OCs, osteoblasts, and osteocytes form the bone multicellular unit and interact via tightly regulated signaling to properly maintain bone homeostasis.

## 3. The Infectious Milieu, a Noxious Brew of Pathogen- and Host-Derived Factors Driving Osteoclastogenesis and Bone Resorption

### 3.1. Pathogen-Derived Factors and Pattern-Recognition Receptors

Many bacteria produce lipoproteins and other products that bind pattern-recognition receptors (PRRs) such as Toll-like receptors (TLRs) [10,11], some of which are present on the plasma membrane of OC lineage cells (Figure 1). *S. aureus* activates TLR2 [12], whereas *P. gingivalis* can activate both TLR2 and TLR4 [13,14]. Other pathogenic products recognized by TLRs include double-stranded ribonucleic acid, heat-shock proteins, and flagellin that activate other TLRs [10,15,16], and may work synergistically [17,18]. Fungi such as *C. albicans* can infect bone and activate TLRs and mannose receptors such as Dectin-1, which are also present on OC precursors [19,20]. Downstream signaling, including activation of MAPKs and NF-κB, shares much in common with RANKL, but the outcome—at least in vitro—is contingent on the state of differentiation of the OC precursors. TLR agonists arrest OC differentiation when acting simultaneously with RANKL on uncommitted progenitors by interfering with NFATc1 induction, but such stimulation following previous RANKL exposure enhances osteoclastogenesis [10]. These effects parallel those of the inflammatory cytokine TNFα on RANKL-induced osteoclastogenesis, described below. In conditions where cell autonomous OC differentiation is inhibited, the primary effect of PRR activation is the production of pro-inflammatory cytokines such as TNFα, interleukin (IL)-1β, and IL-6, which can have paracrine osteoclastogenic effects [21]. The net effect of TLR-activating pathogenic factors in vivo is osteoclast-mediated bone loss in both human patients and animal models.

Another major pathway for microbial products to influence OC differentiation is via inflammasomes, which are very large complexes consisting of a receptor (such as nucleotide-binding oligomerization domain, leucine-rich repeat-containing protein 3 (NLRP3), or absent in melanoma 2 (AIM2)), the adaptor apoptosis-associated speck-like protein containing a CARD (ASC), and caspase-1 [22]. *S. aureus* and *P. gingivalis* can activate both NLRP3 and AIM2 [23,24,25,26], which leads to cleavage of pro-IL-1β, pro-IL-18, and gasdermin-D, the pore-forming unit required for release of mature IL-1β. Inflammasome activation in pre-OCs, either by microbial or host products, promotes OC differentiation and bone resorption in a cell-autonomous manner and contributes to the overall inflammatory milieu [27,28,29,30].

The TLR and inflammasome responses to microbial products are not limited to the OC lineage during bone infection, but rather are activated in most cells, including non-OC myeloid cells and osteoblasts. This leads to high levels of IL-1β and other cytokines such as TNFα, IL-6, and IL-17A that amplify the inflammatory milieu and have osteoclastogenic activity [31,32,33,34,35]. Osteoblasts also respond to microbial products through TLR and inflammasome pathways by upregulating RANKL and downregulating decoy receptor OPG, as well as releasing inflammatory cytokines [36,37,38,39].

Staphylococcal protein A, a molecule important to the microbe’s ability to evade immune clearance, increases RANKL-induced OC differentiation via MAPK and NF-κB pathways [40,41] perhaps by binding to TNFR1, which mediates its effects on osteoblasts [42,43]. Protein A can also form IgG immune complexes that act on the Fc receptor of pre-OCs to promote osteoclastogenesis through a TLR2/MyD88-dependent mechanism that signals to NFATc1 and NF-κB [44]. Without protein A, *S. aureus* does not promote bone loss in mice following intraperitoneal or calvarial inoculation [44]. These in vivo effects are likely both direct and indirect, as this virulence factor causes the release of IL-6 and RANKL from osteoblasts [42,45].

Another *S. aureus* factor, toxic shock syndrome toxin 1 (TSST-1), is a superantigen that functions to inhibit the immune response of host cells. TSST-1 is not cytotoxic, but enhances the bone resorption of mature OCs on mineralized matrix [46]. However, the mechanism for this effect has not been determined. *S. aureus* also makes an array of toxins, including phenol-soluble modulins (PSMs), that kill OCs as effectively as other cells such as monocytes and osteoblasts and are thus unlikely to have direct osteoclastogenic or pro-resorptive effects [46,47]. Nevertheless, loss of PSMα 1/2 reduces bone loss in mice with *S. aureus* bone infections, although the cellular mechanism in vivo, whether via the survival of osteoblasts or the death of OCs, has not been determined [48].

*P. gingivalis* increases the osteoclastogenic potential of OC precursors both locally and systemically [49]. In addition to LPS-mediated TLR activation, *P. gingivalis* releases gingipains, cysteine proteases that can induce osteoclastogenesis of RANKL-treated OCs and enhance OC activity through an increase in integrin αvβ3 and degradation of OPG in co-culture with OBs [50,51]. Phosphoglycerol dihydroceramide, a distinctive ceramide of *P. gingivalis*, has also been found to promote RANKL-induced osteoclastogenesis. This unique lipid does not appear to work via either TLR or inflammasome recognition, but rather by interacting with nonmuscle myosin IIA and activating Rac [52].

### 3.2. Cytokines and Other Host-Generated Osteoclastogenic Factors

The signaling induced by many of the inflammatory cytokines produced during infection, including TNFα, IL-6, IL-1β, and IL-17A, share much in common with RANKL, including NF-κB and MAPKs. These factors enhance the OC differentiation of myeloid progenitors in the presence of RANKL [21,33,35,53]. The most intensively studied has been TNFα, which has potent synergistic effects with RANKL. Although similar to TLR activation, it depends on the relative timing of the stimuli in vitro. Exposure of uncommitted progenitors to TNFα prior to or simultaneous with RANKL prevents OC differentiation, whereas even small amounts of RANKL before TNFα lead to robust osteoclastogenesis [31,54]. Early M-CSF was also discovered to have negative effects on osteoclastogenesis that are mitigated by culture on bone rather than plastic [55]. This calls into question the relevance of several of the in vitro studies on plastic that show inhibitory effects of TNFα and other inflammatory stimuli that seem to contradict the clear osteolytic actions of these cytokines in vivo.

IL-17A is a particularly potent osteoclastogenic cytokine induced by the action of TNFα on T cells. IL-17A stimulates the release of other inflammatory cytokines and RANKL from osteoblasts and synovial fibroblasts in rheumatoid arthritis [56,57,58,59] and is also induced during iOM and septic arthritis [60,61,62]. This factor, produced by the Th17 subset of CD4+ T cells, also induces RANK expression by OC progenitors [63]. TNFα acts on mesenchymal lineage cells to elicit IL-1β expression, which subsequently induces RANKL expression from the same mesenchymal lineage cells [33]. M-CSF, which is also required for osteoclastogenesis, is likewise upregulated by the action of TNFα on stromal cells [64]. Therefore, although several experimental models have demonstrated TNFα-mediated osteoclastogenesis in the absence of RANK or RANKL [35,54,65], their relevance to the context of infection, which induces a plethora of osteoclastogenic stimuli, including RANKL, is unclear.

Infection typically induces necrosis in bone, likely via vascular compromise. Host-derived damage-associated molecular patterns (DAMPs) emanating from dying cells contribute to PRR activation during infection. It was recently demonstrated in sterile models of osteocyte necrosis that macrophage-inducible Ca^2+^-dependent lectin receptor (MINCLE), an ITAM-associated PRR, drives osteoclastogenesis [66]. This mechanism is likely operative in the presence of pathogens as well. Other DAMPs, such as S100 proteins and high-mobility group box 1 (HMGB1), likely activate other PRRs and contribute additional osteolytic stimuli [67]. The bone matrix, in the form of hydroxyapatite particles, activates OCs via the inflammasome [29]. Purine metabolites are also released by osteocyte necrosis and have pro-resorptive activity. Adenosine diphosphate (ADP) promotes the activation of OCs through the P2RY12 receptor [68]. ATP bindsP2X7R, stimulating NF-κB and PKC/Ca^2+^ signaling to induce osteoclastogenesis and Syk to promote resorption [69], as well as P2Y6R, which promotes OC survival [70]. Thus, a wide array of pathogen- and host-derived factors contribute to the recruitment and activation of OCs in the context of iOM (Figure 1).

## 4. Immune Modulation by OCs and Their Conventional and Unconventional Precursors

### 4.1. OC Precursors as Potential Immune Suppressors

OCs differentiate from myeloid progenitors that overlap with cells known as myeloid-derived suppressor cells (MDSCs), a heterogeneous population of immature monocytes and granulocytes known for their immunosuppressive effects [71]. MDSCs are expanded and activated by a number of pro-inflammatory and infectious stimuli and inhibit T cells and innate immune functions of other myeloid cells via release of ROS, NO, and enzymes such as arginase-1 [71]. Several groups have shown that MDSCs can also differentiate into OCs and contribute to bone erosion in various noninfectious models [72,73,74,75], so they likely have the potential to cause tissue damage in iOM (Figure 2). In mouse models of prosthetic-associated *S. aureus* biofilms, MDSCs have been shown to accumulate at higher rates than neutrophils and macrophages, but their role in disease progression is likely complex. With *S. aureus* biofilms, MDSCs are partially responsible for bacterial persistence [76,77,78], whereas in the case of planktonic *S. aureus*, MDSCs can be beneficial in helping resolve acute infection [79]. One study using an in vitro human system of granulocytic MDSC production showed a reproducible biphasic response among many *S. aureus* strains, with low levels of enterotoxins causing MDSC expansion and higher levels inhibiting their proliferation [80]. In *P. gingivalis* infection, several subpopulations of MDSCs were induced and the CD11b^+^Ly6G^+^Ly6C^hi^ monocytic MDSCs were shown to suppress T cell activation [81]. The generation and activation of MDSCs and their subsequent effect on iOM progression may also depend on pathogen-specific virulence factors and their local concentrations. Overall, much remains to be learned about the role of MDSCs in the context of bone infection, as it is not yet clear if they represent an important pool of OC progenitors that mediate tissue damage, whether they impair the immune response and contribute to bacterial persistence or proliferation beyond prosthesis-associated biofilms, or whether they can be beneficial and limit tissue damage without promoting the survival of pathogens.

### 4.2. Relationship between OCs, Dendritic Cells, and T Cells

Although not a predominant pathway during bone homeostasis, several studies have shown that OCs can arise from dendritic cells (DCs) in pathological conditions [82]. Furthermore, transcriptionally, OCs seem to more closely mimic DCs than monocytes [82,83]. The demonstrated ability of OCs to present antigens to both CD4 and CD8 T cells places them firmly in the category of antigen-presenting cells [9], an activity compatible with ongoing bone resorption [84]. Several studies have also shown that T cells in bone marrow are most frequently found near OCs [85,86,87]. Many studies have shown that the priming of T cells that occurs via interaction with OCs preferentially polarizes them toward immunosuppressive regulatory states regardless of whether they express CD4 or CD8 (Treg and T_c_reg, respectively) [84,88,89,90,91] and that these regulatory T cells feed back on OCs to inhibit resorption. However, Ibáñez et al. also identified a minor subset of DC-derived OCs under chronic inflammatory conditions that produce high levels of inflammatory factors that, among other effects, enhance T cell activation [84]. Loss of cathepsin K, a major bone-degrading enzyme released by OCs, inhibits inflammation in an infectious periodontitis model, although the dependence of this effect on OCs was not investigated [92,93]. Overall, the primary role of OCs during infection may be to enhance rather than suppress T cell-mediated inflammation (Figure 2). However, whether the adaptive antimicrobial response is affected by OCs via DC-like antigen presentation is still unclear. Further, it is not clear if the origin of OCs from DCs rather than monocytes in a disease state such as infection impacts their immunoregulatory or resorptive functions.

## 5. Intracellular Infection of OCs Providing a Proliferative Niche

Thus far, we have considered the response of OC lineage cells to pathogenic factors, as well as how this response affects immunity. However, a third way in which the OC can interact with pathogens is through intracellular infection. Several groups have demonstrated intracellular infection of osteoblasts by *S. aureus* (reviewed in [94]) as well as osteocytes [95,96]. Within these mesenchymal lineage cells, the bacteria persist but do not proliferate. Due to the long lifespan of osteoblasts, it has been proposed that these intracellular bacteria represent the seeds of persistent and recurrent infections. Recently, we showed that although OCs and their uncommitted myeloid progenitors take up *S. aureus* similarly, their handling of intracellular bacteria is quite different. OCs are not only unable to kill *S. aureus*, but also allow its intracellular proliferation, albeit with significant heterogeneity between cells [97] (Figure 3). *Mycobacterium tuberculosis*, an obligate intracellular pathogen capable of infecting bone, has also been found to proliferate within OCs to a much higher level than in precursors [98]. Finally, although data in primary cells is lacking, *P. gingivalis* can infect the macrophage cell line RAW-D and enhance its ability to form OCs after RANKL priming [99]. The behavior of *P. gingivalis* during this osteoclastogenic culture was not examined, so it is not clear whether it was able to proliferate. Nevertheless, it appears that intracellular infection of OCs by pathogens is not uncommon and the ability of these cells to restrain proliferation is limited. Furthermore, recent studies of OC fate in vivo point to a much longer lifespan than previously thought [100], leaving open the possibility that OCs may not only provide an intracellular niche for pathogenic expansion during acute infection, but could also represent a long-term safe haven.

## 6. Clinical Implications

Bone infections in patients can be divided into two major categories: those that occur in native bone and those associated with prostheses such as joint-replacement hardware, known as prosthetic joint infections (PJI). Although activation of OCs during bone infection has been well recognized in hematogenous or soft-tissue infection-associated iOM, the problem of bone loss has largely been attributed to osteonecrosis due to a combination of vascular compromise and cytotoxicity. Therefore, targeting the OCs has not been seriously considered as a therapeutic approach. In the case of PJI, the role of neutrophils in the periprosthetic membrane in inducing OCs [101] is similar to the role of macrophages in the aseptic loosening of prostheses [102]. However, the presence of bacterial biofilms, which are resistant to antibiotics, has driven treatment paradigms that require mechanical debridement, replacement of hardware, and antibiotics, but not the specific targeting of other host factors such as OCs. Antiresorptive drugs have been successfully used in sterile forms of osteomyelitis, including chronic non-bacterial osteomyelitis and diffuse sclerosing osteomyelitis [103,104,105], as well as to reduce aseptic loosening of prosthetic joints [106]. These OC-targeting therapeutics have been implicated in cases of osteonecrosis of the jaw (ONJ), which is considered to be an entity separate from iOM of the jaw [107]. Although the pathophysiology of this type of medication-related ONJ remains poorly understood and exposure of bone to oral microflora may be involved [108,109,110], it is not at all clear that pathogens are significant drivers of ONJ, as they are not always detected histologically [111]. Infection of long bones or vertebrae is not listed as a potential complication of antiresorptive therapy [112]. Furthermore, early bisphosphonate treatment can mitigate bone loss in patients with vertebral osteomyelitis without any indications of worsening infection [113]. Therefore, there is no evidence that treatment of iOM patients with anti-OC therapies is likely to be detrimental. If OCs indeed promote both bacterial replication and inflammatory bone loss, it is possible that limiting their induction will be a beneficial adjunct to antibiotics. Notably, that RANKL targeting, such as with denosumab, is likely to be more useful than bisphosphonates since it prevents both osteoclastogenesis and bone resorption.

## 7. Conclusions

Bone infections are associated with increased OC differentiation and OC-mediated bone loss due to the direct actions of factors released by pathogens and the host’s inflammatory response. In addition to the monocytic precursors that predominate during homeostasis, infectious conditions can expand non-traditional pools of OC precursors that include MDSCs and dendritic cells. Besides supplying additional bone-resorbing cells, the functions of these precursors may contribute to the inflammatory milieu or inhibit an adaptive immune response. Mature OCs may also produce cytokines and induce regulatory T cells that modulate the host response to infection. Finally, OCs, which can be long-lived in vivo, may represent an intracellular niche for bacterial persistence or proliferation, providing one more reason that infections in bone are sometimes difficult to eradicate. More research is needed to determine if therapeutic targeting of the OC can reduce the incidence of persistent and recurrent iOM.

## Figures and Tables

**Figure 1 cells-09-02157-f001:**
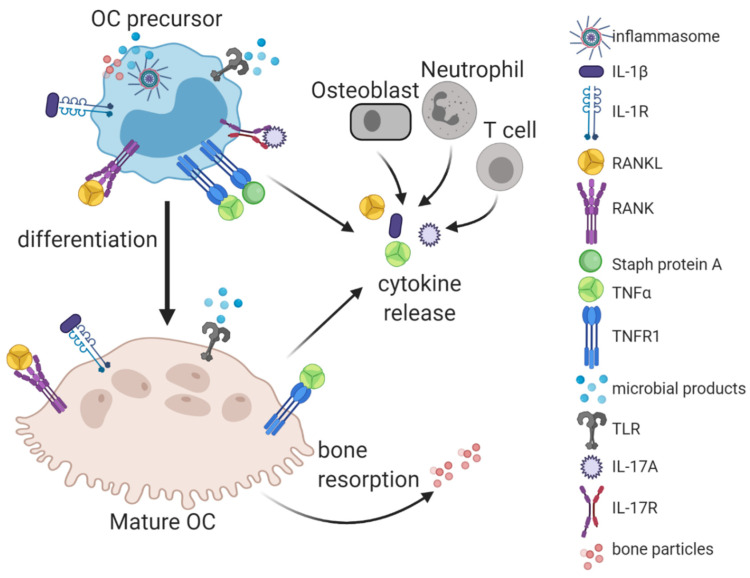
Osteoclastogenic and pro-resorptive factors in bone infection. Both OC precursors and mature OCs have numerous receptors (R), including inflammasomes, IL-1R, RANK, TNFR1, TLRs, and IL-17R, that allow them to respond to factors generated by both pathogens (staphylococcal protein A and other microbial products) and hosts (IL-1β, RANKL, TNFα, and IL-17A). Cytokines are released from many cells in the microenvironment including OC precursors and mature OCs, osteoblasts, neutrophils, and T cells. Bone necrosis and bone resorption can also generate microbial products and bone particles that activate the inflammasome and increase OC differentiation in concert with inflammatory cytokines.

**Figure 2 cells-09-02157-f002:**
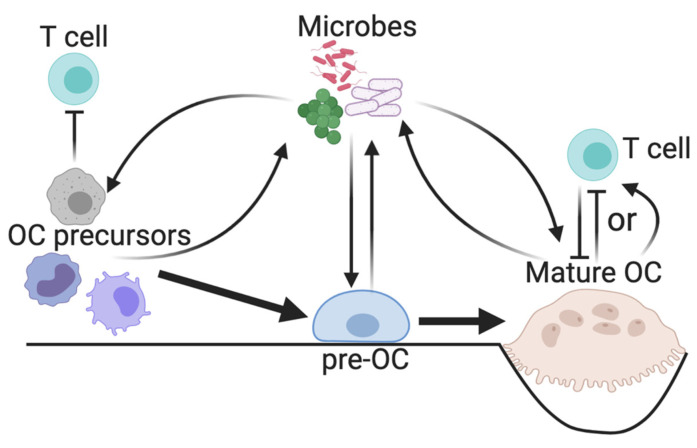
Relationship between microbes, OC lineage cells, and T cells. The presence of microbes in bone expands OC precursors, including the homeostatic monocytic precursors, those with myeloid-derived suppressor cell (MDSC) properties, and dendritic cells (DCs). Both MDSC-type OC precursors and mature OCs can inhibit pro-inflammatory T cell functions, whereas DC-derived OCs have been shown to enhance T cell activation. Whether the primary OC effect on T cells during infection is mainly inhibitory or pro-inflammatory has not yet been definitively demonstrated.

**Figure 3 cells-09-02157-f003:**
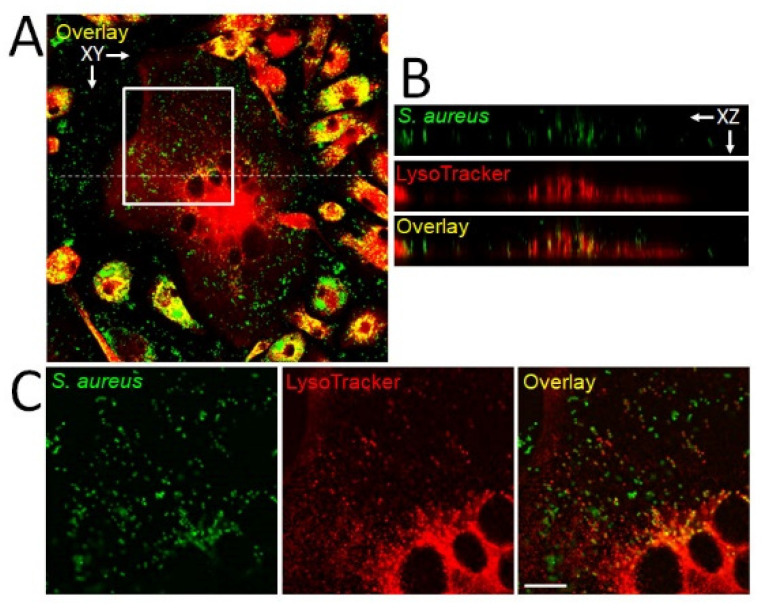
Internalization of *S. aureus* by mature OC and pre-OCs. Murine bone-marrow monocytes were differentiated in RANKL and M-CSF for 3 days, then infected with GFP-labeled *S. aureus* as previously described [97]. After 18 h, cultures were fixed and stained with LysoTracker and imaged using a Nikon A1RSi confocal microscope and Nikon NIS-C Elements software. (**A**) Low-power image of a multinucleated OC surrounded by mononuclear pre-OCs in the XY plane. In this image, the degree of co-localization of bacteria (green) and lysosomes (red), indicated in yellow, is particularly variable in the pre-OCs, but similar variation occurs in OCs as well, as previously shown [97]. (**B**) Images in the XZ plane along the dotted line in (A) show that the bacteria are inside, rather than on top of, the OC. (**C**) XY plane images from the white box in A demonstrate only partial co-localization of bacteria and lysosomes in the mature OC. Scale bar, 10 µm. Microbes that avoid lysosomes are more likely to survive and proliferate than those that end up in these acidified vesicles.

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
