# Peer review of "Multitasking by the OC Lineage during Bone Infection: Bone Resorption, Immune Modulation, and Microbial Niche"

_cells, 2020, doi:10.3390/cells9102157_

Round 1
Reviewer 1 Report
This is a very nice and informative review article depicting the major cellular and molecular events of bone infections. The authors provide a comprehensive description of the role of osteoclasts in this pathological state emphasising on both the response of OCs to inflammation as well as their possible contribution to the proliferation and survival of pathogens. The illustrations that the authors used are explanatory and help the reader to understand briefly the mechanisms underlying host cells and pathogens interactions. The manuscript focuses on an interesting topic with direct clinical implications, covers many aspects of the disease and is well written. Some minor points should be addressed.
- Line 31: necrotic sections or lesions?
- Lines 47-49: Osteoprotegerin must also be mentioned here as an important regulator of osteoclastogenesis which is inextricably linked to the RANK-RANKL-OPG axis and is actively implicated in the osteoblast-osteoclast communication.
- Please clearly state what the abbreviations stand for (e.g. ITAM, HMGB1 etc.).
- Lines 52-54: Mature OCs also express TRAP and specifically the 5b isoenzyme, while TRAP5a is mainly derived from macrophages and dendritic cells. Please comment.
- Line 125: Delete second induced.
- Line 143: Something is missing (TNFα on stromal cells?)
- Lines 203-204: This is vague. If the primary role of OCs is to enhance, rather than suppress, T cell mediated inflammation, it contradicts with what Figure 2 shows as well as what is written in the legend.
- Use italics as appropriate (in vitro, in vivo, et al).
Author Response
Thank you for your positive but careful evaluation of the manuscript. Each of your points has been addressed as follows:
- Line 31: necrotic sections or lesions? We have changed this to “necrotic areas.”
- Lines 47-49: Osteoprotegerin must also be mentioned here as an important regulator of osteoclastogenesis which is inextricably linked to the RANK-RANKL-OPG axis and is actively implicated in the osteoblast-osteoclast communication. Added to lines 47-48.
- Please clearly state what the abbreviations stand for (e.g. ITAM, HMGB1 etc.). All are now stated.
- Lines 52-54: Mature OCs also express TRAP and specifically the 5b isoenzyme, while TRAP5a is mainly derived from macrophages and dendritic cells. Please comment. We have edited the sentence to specify TRAP5b.
- Line 125: Delete second induced. Second instance changed to “produced.”
- Line 143: Something is missing (TNFα on stromal cells?) “on” added.
- Lines 203-204: This is vague. If the primary role of OCs is to enhance, rather than suppress, T cell mediated inflammation, it contradicts with what Figure 2 shows as well as what is written in the legend. We have edited this text as well as modified the figure and legend to clarify our point that both activation and suppression are possible, and although the net result in vivo is inflammation during infection, we do not yet know in this context how this interaction works with certainty.
- Use italics as appropriate (in vitro, in vivo, et al). We have gone through the manuscript to correct these.
Reviewer 2 Report
The manuscript entitles "Multitasking by the OC lineage during bone infection: bone resorption, immune modulation, and microbial niche " by Roper et al. is a very interesting review on the different roles of osteoclasts during bone infection. The reviewer has only a couple of comments to this complete review:
-paragraph 6. Clinical Implications is not very clear and too concise. It should be extented
-Figure 3. Technical details sould be provided (e.g. type of miscope used, image acquisition methods).
Author Response
Thank you for your interest in this topic.
1. Clinical implications section has been expanded, and not also refers to prosthetic joint infections.
2. Technical details have been added to Fig legend 3. More detail can be found in the reference #97.